# Suicide Possibility Scale Detection via Sina Weibo Analytics: Preliminary Results

**DOI:** 10.3390/ijerph20010466

**Published:** 2022-12-27

**Authors:** Yun Gu, Deyuan Chen, Xiaoqian Liu

**Affiliations:** 1School of Electronic, Electrical and Communication Engineering, University of Chinese Academy of Sciences, Beijing 101408, China; 2Institute of Psychology, Chinese Academy of Sciences, Beijing 100101, China; 3Department of Psychology, University of Chinese Academy of Sciences, Beijing 100049, China

**Keywords:** suicidal ideation, machine learning, suicide possibility scale

## Abstract

Suicide, as an increasingly prominent social problem, has attracted widespread social attention in the mental health field. Traditional suicide clinical assessment and risk questionnaires lack timeliness and proactivity, and high-risk groups often conceal their intentions, which is not conducive to early suicide prevention. In this study, we used machine-learning algorithms to extract text features from Sina Weibo data and built a suicide risk-prediction model to predict four dimensions of the Suicide Possibility Scale—hopelessness, suicidal ideation, negative self-evaluation, and hostility—all with model validity of 0.34 or higher. Through this method, we can detect the symptoms of suicidal ideation in a more detailed way and improve the proactiveness and accuracy of suicide risk prevention and control.

## 1. Introduction

Suicide is a conscious act of voluntarily and actively ending one’s life [1]. Despite today’s level of medical health, according to WHO statistics, nearly 1 million people commit suicide each year worldwide. Approximately 287,000 people commit suicide each year in China, accounting for one third of global suicide deaths and the fifth most-common cause of death, making it an important public health and mental health problem in China and globally [2]. However, the identification of suicidal individuals is still an undoubtedly complex and difficult task [3]; therefore, it is important to find an efficient method for suicide detection.

Current technologies for suicide detection span manifold domains and methods. Traditional approaches use clinical methods with patient–clinic interaction [4] to have subjects fill out specific suicide questionnaires and scales and to have experts analyze the contents of the completed assessments to draw the appropriate conclusions. In other words, clinical treatment seeks an understanding of the psychology behind suicidal behavior [5]. Of course, this approach is extremely dependent on the clinician’s own expertise and the face-to-face interaction at the time. The current study uses a judgmental analysis of suicide based on the Suicide Probability Scale (SPS) [6], which is an established and reliable questionnaire for assessing the likelihood of suicide that uses subscale scores and total scores to determine the patient’s likelihood of suicide. However, some at-risk people conceal their true situations [3], and practice effects appear after multiple participation in a given assessment [7].

In 2012, a 22-year-old girl named Zou Fan left her last words on Weibo that she was suffering from depression and decided to die. She hanged herself in her dormitory. The police and medical personnel tried their best to save her life but were unable to do anything. In response to the news, netizens were generally saddened and sorry, and more and more people are calling for attention to depressed people to prevent them from committing suicide. For psychological disorders leading to suicide, the earlier the risk group is detected, the earlier treatment can be provided to avoid the aggravation of suicidal tendencies; however, questionnaires lack timeliness. Moreover, people with suicidal ideation do not tend to fill out questionnaires voluntarily, and questionnaires are not active interventions. Thus, we need more timely and proactive methods to detect suicide. In addition to the traditional method, texts actively generated by risk groups enable better detection [8]. With the advancement of Internet technology, more and more people share their lives, experiences, views, and emotions in those online venues [9], which provides data sources and application platforms for us to monitor suicide.

An increasing number of studies have tried to detect suicide from social media texts [10] and have been able to quickly identify users with suicidal ideation and intervene in a timely manner. Users’ posts on social networking sites reveal a wealth of information and their language preferences. Through exploratory data analysis of user-generated content, we can gain insights into language usage and linguistic cues from suicide attempters. Suicide-related keyword dictionaries and lexicons have been manually built to enable keyword filtering [11] and phrase filtering [12]. Gunn and Lester [5] analyzed the Twitter posts of suicide attempters in the 24 h prior to their deaths. Coppersmith et al. [13] analyzed the language use in data from the same platform. Suicidal thoughts may involve strong negative emotions, anxiety, hopelessness, or other social factors. These thought reactions can be expressed in the text, as some words and phrases include “kill”, “suicide”, “feel alone”, “depressed”, and “cutting myself”. A lexicon based on words such as these can be used as a feature input to identify the possibility of suicide.

The identification of suicidal ideation is mostly based on binary classification models, and suitable feature selection and machine-learning methods have been used to construct classification models. One of the keys is to select the appropriate features. Table 1 shows some of the recent studies on suicide detection based on social media. From the selection of dictionaries, we can find that they [14,15] are based on LIWC dictionaries alone for feature extraction or the fusion of features extracted from Chinese suicide dictionaries [16] and LIWC or construct a new dictionary according to the data situation [17,18] and use it for feature extraction. Most of the modeling methods use machine-learning SVM methods to build discriminative models, but there are also some studies that choose to train neural networks to build models, such as Shing [19] using CNN methods to build models to achieve the desired results. Q. Cong [8] using a variety of modeling methods—including training LSTM networks—to achieve accuracy above 0.9 (although random forest achieved better results in that study).

In summary, the existing studies have been able to detect suicidal ideation to some extent, but there are still some shortcomings and areas for improvement. The traditional questionnaire method and scales can help us identify people at risk with some reliability, but it does not allow timely detection or intervention and requires active completion by the patient, making it highly passive and unable to automatically detect people at risk. In contrast, the automatic detection method can identify suicidal tendencies from text, which allows for a timely and automatic intervention. Therefore, we combine the two methods and use the automatic text detection method to obtain the SPS results. First, for the purpose of the study, most of the models for suicide detection are still based on classification models, and the results are a binary yes or no. However, for suicide early warning and risk judgment, in most cases, absolute judgments cannot be made, and a risk prediction and assessment of suicidal tendency is required to enable targeted treatment plans. Therefore, the results of this model are no longer categorized but modeled and predicted based on Weibo data for each of the four subscale scores of the Suicide Possibility Scale, which is mature and reliable. Experts can diagnose individuals based on the predicted results to obtain a more reliable suicide likelihood analysis. Second, with respect to the research methodology, unlike other studies that combine features for a single or a few lexical dimensions, our study selects six lexical dimensions to obtain more valuable and comprehensive linguistic features and then selects the optimal set of features to build the model. All aspects of the above appear to enable proactive and automated detection of SPS results and provide a new approach to suicide detection.

## 2. Materials and Methods

This study aimed to investigate the correlation between the linguistic features of textual social media content and suicide likelihood ideation and provide a machine-learning approach combining feature screening and linear regression to detect the four indicators of the SPS [6]. The working principle of the realization method of this study is as follows.

First, the subjects’ Weibo text data were collected over a period of time. Then, several dictionaries were used as the basis for language feature extraction, including the Weibo Basic Mood Lexicon [20], the individualism/collectivism lexicon [21], The Chinese Version of the Moral Foundations Dictionary [22], the Moral Motivation Dictionary [23], the Chinese suicide dictionary [24], and Language Inquiry and Word Count (LIWC) [25]. The relationship between the multidimensional features and the model is judged using the Akaike information criterion (AIC) to filter out the best features for the model. The features are then used as model inputs, and a multiple linear regression algorithm is used to construct a suicide likelihood prediction model. Figure 1 illustrates the above modeling process.

### 2.1. Data Collection

The training data for this study were obtained from Sina Weibo, the largest online social media platform in China. We recruited 1062 subjects on our platform and asked them to fill out the Chinese version of the SPS [6], an international questionnaire used to assess suicide attempts in adolescents and adults. The original English version of the questionnaire [26], developed by John G. Cull, Ph.D., and Wayne S. Gill, Ph.D., has good reliability and validity and consists of 36 questions with four dimensions: hopelessness, suicidal ideation, negative self-evaluation, and hostility. The hopelessness dimension consists of 12 questions, the suicidal ideation dimension consists of eight questions, the negative self-evaluation dimension consists of eight questions, and the hostility dimension consists of seven questions. Each question was scored on a 1~4 scale, which can be simply interpreted as the higher the score, the greater the likelihood of suicide. With the subjects’ authorization, a crawler was used to download all their original Weibo data for the 30 days prior to the date the scale was completed. The users were also filtered according to the number of posts in 30 days, and only active users with 10 or more posts were retained. A total of 481 valid samples were obtained, with the average number of posts being 62. The demographic information of the valid samples can be found in Table 2, and the subjects’ scores on the SPS scale can be seen in Table 3.

### 2.2. Feature Extraction

#### 2.2.1. Dictionary

In this paper, we used six dictionaries for linguistic feature extraction. Detailed descriptions of the six dictionaries can be found in Table A1, and the following is a brief description of the dictionaries and how the corresponding dictionaries relate to suicide. including

(1)The Weibo Five Basic Mood Lexicon [20]. It includes an 818-word microblogging basic emotion lexicon constructed by analyzing more than 1.6 million users to measure five basic social emotions (happiness, sadness, disgust, fear, and anger) in microblogs, and verify their validity. Psychological studies have shown that suicide results from a cumulative cause [27] and that the accumulation of negative emotions and repeated outbursts are causative factors of suicide and mediate suicide risk [28].(2)The Individualism/Collectivism Lexicon [21] provides a dictionary of collectivism and individualism based on Weibo data. The data for this dictionary also come from Weibo and shows that individualism has a significant positive relationship with suicide rate [29].(3)Researchers built the Chinese suicide dictionary [24] from 4653 posts on Sina Weibo and two Chinese sentiment dictionaries, and it has good performance in identifying suicide risk on Weibo.Some studies have shown that group-oriented (binding) moral intuitions are associated with lower suicide risk, while individual-oriented (individualizing) moral intuitions are associated with higher suicide risk [30]. Therefore, We have also selected two dictionaries related to morality.(4)The Chinese Version of the Moral Foundations Dictionary [22] is based on moral foundation theories and the Moral Foundations Dictionary of Graham and colleagues (2009) in the Chinese context.(5)The Moral Motivation Dictionary [23] was initially designed by Frimer (2013) for LIWC. It contains 349 words in the agency dimension (e.g., accomplish, defeat, spent) and 146 words in the communion dimension (e.g., accepting, care, kindness).(6)The SCLIWC, the Simplified Chinese Microblog Word Count tool [31], is a Chinese version of the classic and commonly used LIWC [25] tool for sentiment analysis, allowing a one-stop solution from automatic Chinese word segmentation to psycholinguistic analysis.

#### 2.2.2. The Processing

We will use LIWC as an example to introduce how to extract features from the original text based on the corresponding dictionary. The LIWC program has two central features—the processing component and the dictionaries. The processing feature is the program itself, which opens a series of text files—which can be essays, poems, blogs, novels, and so on—and then goes through each file word by word. Each word in a given text file is compared with the dictionary file [25].

For example, if LIWC were analyzing this sentence:

It was a terrible thing.

The program would first look at the word “it” and then see if “it” was in the dictionary. It is and is coded as a function word, a pronoun, and, more specifically, an impersonal pronoun. All these LIWC categories would then be incremented. Next, the word “was” would be checked and would be found to be associated with the categories of verbs, auxiliary verbs, and past tense verbs. After going through all the words in the text, LIWC would calculate the percentage of each LIWC category. Therefore, for example, we might discover that 2.34% of all the words in a given book were impersonal pronouns and 3.33% were auxiliary verbs. The LIWC output, then, lists all LIWC categories and the rates that each category was used in the given text.

The multidimensional linguistic features were extracted from the post contents of 481 microblog users using the above six dictionaries. They were manually filtered, the full 0-dimensional and irrelevant features were screened out, and a total of 121-dimensional linguistic features were obtained.

### 2.3. Model Construction with Machine-Learning Algorithms

By comparing various algorithms, we chose the multiple linear regression algorithm as the machine-learning algorithm for the model. For the multiple linear regression algorithm, we used a stepwise regression approach to select a more appropriate set of independent variables.

The basic idea of stepwise regression is to introduce variables into the model one by one, perform an F test after each explanatory variable is introduced, and perform a *t* test on each of the explanatory variables that have been selected. If the explanatory variable introduced first is no longer significant due to the introduction of subsequent explanatory variables, it is removed. This is undertaken to ensure that only significant variables are included in the regression equation before each new variable is introduced. This is an iterative testing process until neither significant explanatory variables are selected into the regression equation nor insignificant explanatory variables are removed from the regression equation, indicating that the optimal set of explanatory variables is reached at that point. Here, we choose the AIC as the criterion for independent variable selection, and when the AIC reaches a minimum, we obtain the most available set of independent variables. The basic principles of the AIC are as follows:(1)$$AIC=-2lnL({\hat{\theta }}_{L},x)+2p$$
where ${\hat{\theta }}_{L}$ is the maximum likelihood estimate of θ and *p* is the number of unknown parameters. Stepwise regression is performed for each of the four predicted SPS indicators to obtain the corresponding optimal set of independent variables.

Multiple linear regression analysis is a regression analysis conditional on the given values of multiple explanatory variables and is a method to study the linear relationship between a dependent variable and multiple independent variables. In fact, a phenomenon is often associated with more than one factor, and the optimal combination of multiple independent variables to predict or estimate the dependent variable is more effective and better suited to the needs of our experiments than using only one independent variable for prediction or estimation. Moreover, using regression models, the results that can be calculated by standard statistical methods are unique as long as the models and data used are the same.

The general form of a multiple linear regression model is:(2)$$Y=\beta_{0}+\beta_{1}x_{1}+\beta_{2}x_{2}+\beta_{3}x_{3}+\cdot \cdot \cdot +\beta_{j}x_{j}+\cdot \cdot \cdot +\beta_{k}x_{k}+\mu$$
where *k* is the number of explanatory variables, $\beta_{j}\left(j=1,2,\ldots ,k\right)$ is the regression coefficient, and μ is the random error after removing the effect of *k* independent variables on *Y*.

### 2.4. Measures of Model Performance

The performances of the constructed suicide probability detection models were evaluated using reliability and validity. Reliability assesses the consistency of a measure, and validity assesses its accuracy.

We used five-fold cross-validation to calculate the Pearson correlation coefficient of the model to verify the validity of the model. The Pearson correlation coefficient is the test statistic that measures the statistical relationship or association between two continuous variables. It is known to be the best method of measuring the association between variables of interest because it is based on the method of covariance. It gives the correlation between linguistic features of textual social media content and suicide likelihood ideation, and it can also demonstrate the reliability of the model. The basic principles of the Pearson correlation coefficient are as follows:(3)$$\rho_{xy}=\frac{Cov(x,y)}{\sigma_{x}\sigma_{y}}$$
where Cov(x,y) denotes the covariance of the sample and $\sigma_{x}$ and $\sigma_{y}$ denote the standard deviation of the sample.

We calculate the odd-even split reliability of the model. We divide the content of each user’s Weibo posts in half according to odd and even, extract features from the odd and even parts, and then build a model according to the same modeling method to predict the four indicators of SPS [6]. Finally, we compare the correlation between the two sets of prediction data obtained to verify the reliability of the model.

## 3. Results

### 3.1. Feature Extraction Results

By means of stepwise regression, for each subscale dimension, we obtain the corresponding optimal set of features. For different dictionaries of linguistic features, the features selected for each dimension vary and are roughly distributed as shown in Table 4:

### 3.2. Split-Half Reliability

After stepwise regression, the extracted multidimensional language model features were filtered to obtain the optimal feature set. The model was constructed by the machine-learning method of multiple regression. A portion of the data is randomly selected to train the new model and split the remaining data in half according to odd and even numbers as a test set, obtain the corresponding predicted values, and compare the correlation between the two to obtain the corresponding model reliability. Table 5 shows the reliability of the model’s predictions for the four indicators of the SPS scale and the total scores.

### 3.3. Criterion Validity

The Pearson correlation coefficient between the predicted and actual scores of each subscale was calculated using a five-fold cross-validation method, which led to the criterion validity analysis (as shown in Table 5). The results showed that the correlation coefficients reached a significant level, which implies that the developed model has high criterion validity.

## 4. Discussion

The reliability and validity results show that the machine-learning method based on multiple linear regression better predicts the Suicide Possibility Scale and provides a new possibility for the prediction of suicide possibility through non-subjective social media. It is worth noting that the predictive validities of all four indicators of the scale were above 0.34, and the validity of the model for the total score of the scale reached 0.35, indicating that it has great significance in predicting the possibility of suicide through the Suicide Possibility Scale. By looking at the feature extraction results, we find that among the six dictionaries used, SCLIWC, the Moral Foundations Dictionary, and the Chinese suicide dictionary have the most extracted features, indicating that they provide the more dominant and important feature dimensions for accurate prediction of SPS. The reliability results show that suicidal ideation has the lowest reliability results.

The experimental results presented in Table 4 shows that among the set of features extracted based on six dictionaries, three dictionaries retained more features after the stepwise regression of feature filtering, among which SCLIWC contributed the largest share of features. In the case of research on suicide, related texts typically entail the use of LIWC [32], which is a tool for the statistical analysis of corpora using a wide set of dictionaries. Using this tool has become standard in psychological studies on language [33], particularly studies on the language of suicide victims [13,34,35]. The information it provides is also often used in machine-learning algorithms [19,36]. LIWC provides a wide range of linguistic category annotations on the text. Michal Ptaszynski [32] found that the analysis of the obtained LIWC study results enabled several valuable insights into the vocabulary used by suicidal users in comparison to that used by non-suicidal users. Therefore, using LIWC categories as additional features helps the model acquire more important features.

The Moral Foundations Dictionary is another source dictionary with a relatively large number of features [22]. This dictionary is based on the LIWC, which is extended and compressed according to the five dimensions of the appeal. The authors developed several ways to measure people’s use of five sets of moral intuitions: harm/care, fairness/reciprocity, ingroup/loyalty, authority/respect, and purity/sanctity. We can find the category harm present in the set of features of each subscale, which contain a large number of negative words related to harm, while some studies have shown that suicide victims seem to show more self-concern and more negative expressions; use more cognitively exclusive, death-related, and religion-related words; and use fewer work-related words [37]. Therefore, moral-based dictionaries can extract different linguistic features of suicidal and non-suicidal people from Weibo and help in the construction of suicide likelihood models. From Table 4, we can see the Chinese suicide dictionary also contributes important features [24]; it selects initial words from 4653 posts published on Sina Weibo and two Chinese sentiment dictionaries (HowNet and NTUSD), and its performance in identifying suicide risk on Weibo has been confirmed. Its dictionary catalog contains suicide ideation, hopeless, self-regulation, and hostility, which correspond to the four subscale dimensions of the SPS, thus helping us extract the corresponding features more precisely.

By observing the reliability and validity test results of the suicide possibility identification model shown in Table 5, we find that the reliability of suicidal ideation is the lowest. This is due to suicidal ideation having been conceptualized and measured as a state rather than a trait, which would be expected to fluctuate in intensity over time [38]. For a fluctuating state quantity, it is difficult to make the results of each prediction reach a stable state, so there is low reliability.

## 5. Conclusions

As suicide is an increasingly prominent social issue, the traditional questionnaire method, SPS [6], does not allow for automatic detection or timely intervention, and at-risk individuals often conceal their intentions, which is not conducive to early suicide prevention. There is an urgent need for suicide risk markers that do not rely on self-reports, and this study provides a risk warning of suicide likelihood through social media comments on Weibo. Unlike conventional research methods that use machine learning to directly model suicide classification or traditional scale questionnaire analysis, this study combines machine learning and scale questionnaires, and instead of simply constructing a classification model, four-dimensional subscales of the SPS [6] scale for suicide likelihood discrimination are used for predictive analysis, enabling the results of SPS [6] to be automatically obtained and more timely interventions for risk groups. This study also provides a new way of thinking about existing research methods.

This study has some limitations. First, the amount of data in this experiment was not sufficient, and the time span was short. These factors may affect the reliability to some extent, and by observing the demographic informatics background shown in Table 2, we found that most of the subjects were 20–30 years old, which may have led to an age sampling bias. In future studies, subjects with more balanced demographic backgrounds should be enrolled, and expanding the time span of the data can allow the model to cover a wider range of subjects. Second, in addition to the original text data of users, social media also contains a large amount of interactive communication information among users, which can also enrich the dimension of features and help the model to better analyze the psychological condition of users, but we were not able to obtain more information this time. In the future, we will consider introducing more aspects of data to help build the model.

## Figures and Tables

**Figure 1 ijerph-20-00466-f001:**
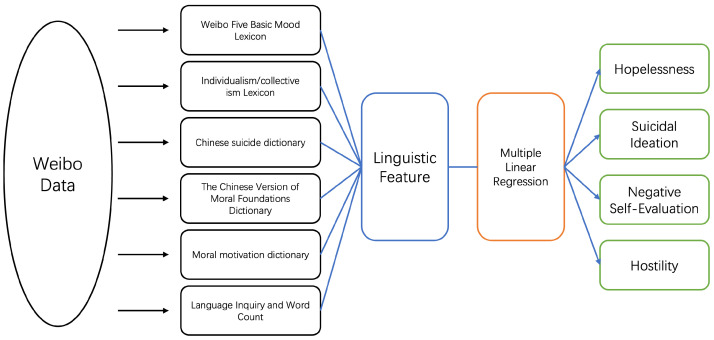
The Model ConstructionMethod.

**Table 1 ijerph-20-00466-t001:** A review of different suicide ideation detection studies on social media.

No	Features Extracted	Modeling Methods	Social Media	References
1	Custom Vocabulary Dictionary	Naive Bayes	Twitter	[17]
2	TFIDF	SVMs	Twitter	[10]
3	Custom Emotion-labeled	Logistic Regression	Twitter	[18]
4	SC-LIWC	SVM	Weibo	[14]
5	Data-driven Dictionary, LIWC, Chinese Suicide Dictionary	SVM, DT	Weibo	[16]
6	TFIDF, LIWC, Sentiment Analysis	Logistic Regression, Random Forest, SVR	Reddit	[15]
7	TF-IDF, LIWC, POS	Random Forest LSTM etc.	Reddit and Twitter	[8]
8	BoWs, empath, readability, LIWC, NRC, LDA etc.	CNN	Reddit	[19]

**Table 2 ijerph-20-00466-t002:** Demographics of participants.

Demographics	Subclass	All Weibo Posts (N = 37,474), n (%)
Gender	Male	9775 (26.08)
	Female	27,699 (73.92)
Regions	country	192 (0.51)
	Town	2540 (6.78)
	Prefecture-level city	12,991 (34.67)
	municipality	21,751 (58.04)
Ages	14~20	5744 (15.33)
	20~30	27,624 (73.72)
	30~52	4106 (10.96)
Profession	Laborer	371 (0.99)
	Civil Servant	883 (2.36)
	Military/Police	32 (0.09)
	Researcher/Teacher	4749 (12.67)
	Media Practitioners	1300 (3.47)
	Unemployed	1153 (3.08)
	Student	16,249 (43.46)
	Doctor/Nurse	1111 (2.96)
	Staff	6895 (18.40)
	Freelancers	1473 (3.93)
	Self-Employed	569 (1.52)
	Other	2689 (7.18)

**Table 3 ijerph-20-00466-t003:** Suicide possibility scale scores.

Dimensions	Average	Standard Deviation	Highest	Lowest
Hopelessness	24.53	4.71	39	12
Suicidal Ideation	11.53	3.27	26	8
Negative Self-Evaluation	20.36	4.31	36	9
Hostility	12.91	2.35	21	7
Total Score	69.32	11.82	106	43

**Table 4 ijerph-20-00466-t004:** Feature Distribution.

Dictionary	Hopelessness	Suicidal Ideation	Negative Self-Evaluation	Hostility
SCLIWC [31]	21	12	18	21
Moral Foundations Dictionary [22]	4	3	4	6
Chinese suicide dictionary [24]	4	3	4	3
Weibo Five Basic Mood Lexicon [20]	1	1	1	1
Individualism/Collectivism Lexicon [21]	1	0	0	1
Moral Motivation Dictionary [23]	1	0	0	0
Total	32	27	19	32

**Table 5 ijerph-20-00466-t005:** Reliability and validity test results of suicide possibility identification model based on microblog text analysis.

Dimensions	Validity	Reliability
Hopelessness	0.34	0.72
Suicidal Ideation	0.35	0.38
Negative Self-Evaluation	0.35	0.47
Hostility	0.36	0.81
Total Score	0.35	0.65

## Data Availability

To protect the participants’ privacy, the original posts used for the analysis are not publicly available but from the corresponding author at a reasonable request.

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
