# Peer review of "Suicide Possibility Scale Detection via Sina Weibo Analytics: Preliminary Results"

_ijerph, 2022, doi:10.3390/ijerph20010466_

Round 1

Reviewer 1 Report

I found this manuscript to be engaging and novel. It is a reasonable attempt to ascertain suicidality via machine learning on social media users as a means of early suicide prevention. The authors did sound research/analysis with methods sufficiently described for repetition. In doing so they clearly stated the limitations of this research, one of the biggest being the small sample size (especially after the exclusion of not that active users). Nevertheless, it seems sufficient for preliminary findings. The other limitation which was not clearly stated was the fact the authors included solely Chinese users (due to the nature of the social media used). That basically means their results are difficult to apply to other languages (users). From the psychiatric perspective, suicidality is a very complex phenomenon which questionnaires and scales only partially address (psychiatric interviews being the gold standard). However, the authors did a sound job of trying to improve and speed up the process that could lead to a psychiatric referral, not trying to "diagnose suicidality" but to detect what could be suicidal thoughts and so on. I am aware that mass analysis of user data (quite possibly without user consent) raises a myriad of ethical questions, but that is out of the scope of this manuscript, nor did the authors get caught in that web (as users consented to this research).

All being said I recommend accepting this manuscript for publication, with a slight change of the title adding "preliminary results" to it.

Reviewer 2 Report

1. The paper is not really talking about social-media analytics, while the authors only focus on Sina Weibo text content analysis. Therefore, the topic sounds too huge to reflect the real range of this study. 

Fine-tune the title or add a suitable subtitle if the paper will be re-submitted in future.

2. It seems that Sina Weibo text content analysis is only some extended methodology of traditional clinical assessment and questionnaire in this study. The authors have paid very little attention on digital and interactive features of social media.

What is the usage scenario of this detection model? Put a Weibo ID request in questionnaire or clinical assessment? How to make a proactive detection effectively happen via social media even if the constructed model is proved to be valid and reliable by yourself?

The paper’s introduction and research design are both in lack of the analysis of social media, ignoring the latest status of Weibo content, the importance of social relationship, the users’ behavior and how they post, share and interact with others.

By variety of well-known reports, most Weibo users prefer to share and forward contents with or without comments. It’s not so frequent for Weibo users to generate original posts daily, weekly or even monthly, especially in lack of text content with the popularity of multi-media content such as pictures, photos and short-video.

It is reasonable this study wants to focus only on text content considering the feasibility of text analysis based on related dictionaries. However, the study shouldn’t just care about the text, the dictionary, the model, the equation while paying so little attention on social media, user behavior, content features, research backgrounder and usage scenario.

After all, the authors have to clarify some key elements of a study. What’s the research questions? What’s the research objectives? What’s the research values? What’s the research innovation and contribution compared with similar studies?

To construct a model? To testify text from Weibo based on some dictionaries? To leverage machine learning algorithms, and for what?

3. Furthermore, what is the value and the contribution of this study? The authors introduced very few details about relevant or similar studies, models and methodologies. There’re many social-media analytics experiments and applications ongoing in Twitter, Facebook and even Sina Weibo from perspectives of marketing, healthcare, governance, security and so on. Think about it once again, what are the differences and innovation comparing with other similar studies?

This is actually a text-analysis-based study but the authors talked less about text details in dictionaries and machine learning extraction or fusion results. The paper has paid too much attention on model design, model constructing and model test. It’s strongly recommended that the paper clarify the research objectives and the research values from introduction to conclusions. Appendixes or tables for more text details about the dictionaries and the machine learning results are also needed.

4. The authors make a simple text analysis so complex. What is the meaning of so-called machine learning and what is the contribution of machine learning? What is the difference between machine learning algorithms and traditional data analytics methods?

How could a model based on machine learning and effective algorithms improve classical dictionaries and content analytics? The model you constructed is from Weibo data collection, feature extraction, reliability and validity criterion, but without any effectiveness verification back to Weibo for new content and more case studies.

Finally, suggest the authors re-think the relationship between the scientific experiment and the qualified paper writing. Maybe the paper will look better if this study going on further with questions listed above could be solved soundly.

5. AND, more basic writing problems as below represent the lack of quality requirement and serious attitude.

Line 208-209   What’s that?

Line 131   I or We?

……

Round 2

Reviewer 2 Report

Wish the authors continue in-depth research and share more valuable findings in future.